# Passive Immunization of Chickens with Anti-Enterobactin Egg Yolk Powder for *Campylobacter* Control

**DOI:** 10.3390/vaccines9060569

**Published:** 2021-06-01

**Authors:** Huiwen Wang, Ximin Zeng, Liu Cao, Qiang He, Jun Lin

**Affiliations:** 1Department of Animal Science, The University of Tennessee, Knoxville, TN 37996, USA; hwang83@vols.utk.edu (H.W.); xzeng3@utk.edu (X.Z.); lcao5@vols.utk.edu (L.C.); 2Department of Civil and Environmental Engineering, The University of Tennessee, Knoxville, TN 37996, USA; qianghe@utk.edu

**Keywords:** anti-enterobactin egg yolk antibody, passive immunity, *Campylobacter jejuni*, chicken, food safety

## Abstract

Enterobactin (Ent) is a highly conserved and important siderophore for the growth of many Gram-negative bacterial pathogens. Therefore, targeting Ent for developing innovative intervention strategies has attracted substantial research interest in recent years. Recently, we developed a novel Ent conjugate vaccine that has been demonstrated to be effective for controlling Gram-negative pathogens using both in vitro and in vivosystems. In particular, active immunization of chickens with the Ent conjugate vaccine elicited strong immune responses and significantly reduced intestinal colonization of *Campylobacter jejuni*, the leading foodborne bacterial pathogen. Given that hyperimmune egg yolk immunoglobulin Y (IgY) has been increasingly recognized as a promising and practical non-antibiotic approach for passive immune protection against pathogens in livestock, in this study, we assessed the efficacy of oral administration of broiler chickens with the anti-Ent hyperimmune egg yolk powder to control *C. jejuni* colonization in the intestine. However, supplementation of feed with 2% (*w*/*w*) of anti-Ent egg yolk powder failed to reduce *C. jejuni* colonization when compared to the control group. Consistent with this finding, the ELISA titers of the specific IgY in cecum, ileum, duodenum, gizzard, and serum contents were similar between the two groups throughout the trial. Chicken intestinal microbiota also did not change in response to the egg yolk powder treatment. Subsequently, to examine ex vivo stability of the egg yolk IgY, the chicken gizzard and duodenum contents from two independent sources were spiked with the egg yolk antibodies, incubated at 42 °C for different lengths of time, and subjected to ELISA analysis. The specific IgY titers were dramatically decreased in gizzard contents (up to 2048-fold) but were not changed in duodenum contents. Collectively, oral administration of broiler chickens with the anti-Ent egg yolk powder failed to confer protection against intestinal colonization of *C. jejuni*, which was due to instability of the IgY in gizzard contents as demonstrated by both in vivo and ex vivo evidence.

## 1. Introduction

A hen’s egg contains a large quantity of egg yolk antibodies (immunoglobulin Y, IgY), which are transferred from serum to yolk during egg formation. The vertically transferred antibodies can provide passive immune protection for the chick, either at the embryonic or post-hatching stages, against pathogens [1]. In recent decades, egg yolk IgY has drawn considerable interest as a passive immune agent and a promising alternative to antibiotics due to several unique features [2,3]. A hen can produce 15–30 g of egg yolk IgY yearly with 2–10% being antigen-specific. Egg yolk IgY is fairly stable under a wide pH range (3.5–11) and high temperatures (up to 70 °C), making it feasible for storage and processing as a feed supplement for food animals. Finally, large-scale maintenance of hens for mass production of desired egg yolk IgY is cost-effective, and collection of egg yolk IgY—a non-invasive practice—is favorable to animal welfare. Consequently, numerous studies have been conducted to develop antigen-specific egg yolk IgY for passive immune protection of food animals, aquaculture, and humans against diverse microbial pathogens, showing partial successes of this approach [1,2]. Therefore, developing pathogen-specific egg yolk IgY is a promising approach to prevent and control microbial infections in food animals for enhanced animal health and food safety [1,3].

*Campylobacter* is the leading bacterial cause of human gastroenteritis worldwide. Campylobacteriosis is responsible for 400–500 million cases of diarrhea each year; the mortality is estimated to be 24 deaths per 10,000 culture-confirmed cases [4].This organism is prevalent in food animals, especially in poultry, posing a significant risk of human infections from farm to fork [5]. Therefore, effective control of *Campylobacter* in poultry, the primary animal reservoir, would reduce the risk of human exposure to this pathogen and have a significant impact on food safety and public health [6]. *Campylobacter* colonization in poultry is determined by the factors involved in complex bacterium-host interaction, which generally include bacterial metabolism, virulence, and stress response, as well as chicken immune system and intestinal microbiota, and have yet to be fully elucidated [7]. Over the past decades, the increasing antibiotic resistance due to extensive use of antibiotics in livestock has raised an urgent need to develop non-antibiotic approaches, such as vaccines and probiotics [5,8], to control foodborne pathogens, including *Campylobacter*. However, these efforts have only gained limited success; developing novel and practical alternative approaches for *Campylobacter* control in poultry is still highly warranted [5].

Recently, we developed a unique conjugate vaccine targeting enterobactin (Ent) [9], a small siderophore compound utilized by different Gram-negative pathogens, including *Campylobacter*, for in vivo growth and colonization [10,11]. The anti-Ent antibodies induced by the Ent conjugate vaccine inhibited in vitro growth of diverse *Campylobacter* strains [9,12]. More importantly, intramuscular immunization of chickens with the Ent conjugate vaccine elicited high levels of systemic anti-Ent IgY, leading to an approximately 3–4 log_10_ unit reduction of *C. jejuni* in the intestine, which was consistently observed in two independent trials [13]. However, despite the significant protection of the Ent conjugate vaccine demonstrated in this proof-of-concept study [13], it was recognized that using an injectable vaccine for *Campylobacter* control in the poultry production system is not cost-effective. To address this issue, we recently optimized conditions for the production of anti-Ent hyperimmune egg yolk IgY using the same Ent conjugate vaccine [14]. We hypothesized that supplementation of chicken feed with the anti-Ent egg yolk powder would control *C. jejuni* colonization in the chicken intestine. This hypothesis was tested in this study using a well-established chicken model of *C. jejuni* infection. Notably, this study, for the first time, assessed the availability and stability of the hyperimmune egg yolk IgY in different segments of the chicken gastrointestinal (GI) tract using both in vivoand ex vivo systems.

## 2. Materials and Methods

### 2.1. Preparation of the Egg-Yolk-Supplemented Chicken Feed

Large quantities of a high titer of anti-Ent hyperimmune egg yolks were generated using an Ent conjugate vaccine as detailed in our recent study [14]. In this work, the hyperimmune egg yolks were further lyophilized into powder using a freeze dryer (VirTis AdVantage Plus EL-85 Benchtop Lyophilizer, SP Scientific Inc., Gardiner, NY, USA) following a published protocol [15] with slight modifications. Briefly, the anti-Ent hyperimmune egg yolks were pooled, diluted with about half volume of sterile PBS, and homogenized using a sterilized egg whisk. The mixture was then transferred into trays with about 500 mL per tray. The trays were wrapped in aluminum foil with several small pores for evaporation and put in a −80 °C freezer overnight for complete solidification of egg yolks prior to lyophilization. Subsequently, the egg yolk mixture in trays was processed in a lyophilizer for about 72 h. Lastly, the lyophilized egg yolk was pulverized into rough powder and stored at −20 °C until use. Prior to the chicken trial, the feed was supplemented with 2% (*w*/*w*) of the freeze-dried egg yolk powder; the dosage of egg yolk powder supplemented in the feed (2%) was in the upper range of most previous animal trials (from 0.04% to 2%) [16,17,18]. The unused egg-yolk-powder-supplemented feed was routinely stored at 4 °C in the chicken facility.

### 2.2. Chicken Trial

The chicken experiment was approved by the Institutional Animal Care and Use Committee at The University of Tennessee, Knoxville, with protocol number 1387–0219. Briefly, 30 newly hatched broiler chicks were assigned into two groups that received regular feed (control group, *n* = 15) or the anti-Ent egg-yolk-powder-supplemented feed (treatment group, *n* = 15) for 14 days. Fresh feed was replenished every 1–2 days. At 5 days of age, 5 chicks in each group were removed for blood sample collection and then euthanized for collecting GI contents (cecum and ileum); the remaining chicks in each group were challenged with 1.5 × 10^6^ colony forming units (CFU) of *C. jejuni* NCTC 11168 via oral gavage. At 9 and 14 days of age (i.e., 4 and 9 days post-infection (DPI)), 5 chicks per group were subjected to collection of blood and various GI contents (cecum, ileum, duodenum, and gizzard). Each of cecum and ileum contents was weighed and serially diluted in Mueller–Hinton (MH) broth (Difco, Sparks, MD, USA). The diluted samples were plated on MH agar plates supplemented with *Campylobacter*-specific selective supplements (Oxoid, Hampshire, UK) and the agar plates were incubated in a microaerobic chamber (5% O_2_, 10% CO_2_, and 85% N_2_) at 42 °C for 2 days. Following incubation, the *C. jejuni* colonies were enumerated and the log_10_-transformed CFU per gram of intestinal contents were calculated to reflect *C. jejuni* colonization level. The other samples were stored at −80 °C prior to immunological and molecular analyses as described below.

### 2.3. Enzyme-Linked Immunosorbent Assay (ELISA)

Indirect ELISA analysis was conducted to determine the specific IgY titer as described previously [9]. Briefly, the ELISA plates (Nunc MaxiSorp, Thermo Fisher, Waltham, MA, USA) were coated with 100 μL of Ent conjugate (300 ng/mL) in a coating buffer (bicarbonate/carbonate coating buffer, pH 9.6) overnight at room temperature. The coated plates were blocked with a blocking buffer (PBS + 0.05% Tween 20 + 5% skim milk) for 1 h at room temperature. Unless specified, each solid sample (feed and GI contents) was gently mixed with saline and supernatant was collected after centrifugation at 5000× *g* for 2 min. The supernatant samples as well as blood samples were 2-fold serially diluted in blocking buffer and incubated with the coated plates for 1 h at room temperature. Following washing procedure (PBS + 0.05% Tween 20), a horse-radish peroxidase-conjugated goat anti-chicken IgY secondary antibody (SeraCare, Milford, MA, USA) was added (2000-fold diluted in blocking buffer) and the plates were incubated at room temperature for 1 h, followed by washing again. The plates were finally developed using an ABTS Peroxidase Substrate Kit (SeraCare) and the reaction was stopped after 30 min by adding 100 μL of stop solution (1% SDS). The absorbance was measured at an optical density of 405 nm (OD_405_) using a microplate reader (BioTek Instruments, Winooski, VT, USA) and data were collected using the Gen5 software (BioTek Instruments). The wells without the addition of the primary antibody served as background control. Endpoint titer was defined as the last dilution at which OD_405_ of the sample wells exceeded the cutoff value (0.1). Duplicate measurements were performed for each sample.

### 2.4. Intestinal Microbiota Analysis

Genomic DNA was extracted from cecum or ileum contents using a QIAamp PowerFecal Pro DNA Kit (QIAGEN, Germantown, MD, USA). The purified genomic DNA was used as the template for PCR amplification targeting the V4 region of the 16S rRNA gene and using the barcoded primer pair of 515F (GTGCCAGCMGCCGCGGTAA) and 806R (GGACTACHVGGGTWTCTAAT) [19]. The PCR was performed in a 25 μL cocktail mix containing 12.5 μL Phusion Flash PCR Master Mix (Thermo Scientific, Waltham, MA, USA), 8.5 μL ultrapure water, 1 μL forward primer (10 μM), 1 μL reverse primer (10 μM), and 2 μL (100 to 150 ng) DNA template. The PCR program included initial denaturation at 94 °C for 3 min, followed by 35 cycles at 94 °C for 45 s, 55 °C for 60 s, and 72 °C for 90 s, with a final extension at 72 °C for 10 min. After amplification, the PCR products were purified using a ChargeSwitch™ PCR Clean-Up Kit (Invitrogen, Carlsbad, CA, USA) and quantified using a NanoDrop spectrophotometer (Thermo Fisher, Waltham, MA, USA). Subsequently, the purified products were pooled together in equal concentrations and further quantified using a KAPA Library Quantification Universal Kit (Roche, Indianapolis, IN, USA). Before being loaded to a MiSeq Reagent Kit v3 (Illumina, San Diego, CA, USA), the library was denatured, diluted, and spiked with PhiX library as an internal control according to the protocol for 16S Metagenomic Sequencing Library Preparation (Illumina, San Diego, CA, USA) [20]. Sequencing of the amplicon libraries was performed with an Illumina MiSeq System.

After sequencing operations, the acquired amplicon sequence data were analyzed in QIIME 2 using the DADA2 pipeline and scikit-learn classifier with reference database SILVA release 138 [21,22,23]. Principal coordinate analysis (PCoA) on microbiota data was conducted and visualized using the R package phyloseq [24]. The raw sequencing data are publicly available in the NCBI Sequence Read Archive with BioSample accession numbers from SAMN18699372 to SAMN18699431 under BioProject ID PRJNA721092 (https://www.ncbi.nlm.nih.gov/sra/?term=PRJNA721092).

### 2.5. Ex Vivo Stability of the Egg Yolk IgY

The GI contents obtained from two independent sources of broilers were used for the evaluation of the ex vivo stability of the egg yolk IgY. The freeze-dried anti-Ent egg yolk powder was dissolved in saline at a ratio of 1:3 (*w*/*w*), followed by centrifugation at 13,000× *g* for 2 min. The supernatant containing the specific egg yolk IgY was used to spike various samples to evaluate the ex vivo stability of the specific IgY as described below.

In the first assay, duodenum and gizzard contents were obtained from the euthanized broilers (7 weeks old) in a slaughter plant in Mississippi (courtesy of Dr. Li Zhang, Mississippi State University); each sample was comprised of pooled contents from 5 randomly selected chickens. The samples were immediately packed with ice packs and shipped to our laboratory via overnight express carrier. Upon receiving the samples, the duodenum contents were gently mixed with 1/3 volume of sterile saline and centrifuged at 5000× *g* for 5 min. The supernatant was spiked with the aforementioned egg yolk IgY solution at a ratio of 7:1 (*v*:*v*), gently mixed, and incubated at 42 °C (chicken body temperature) for 2.5 h; the sample mixed with saline served as a control. The gizzard contents were gently mixed with the same volume of saline and centrifuged at 5000× *g* for 5 min. The supernatant was collected and the pH of aliquot was adjusted to either 3.0 or 7.0 using less than 2% volume of HCl or NaOH, respectively. Subsequently, the pH-adjusted gizzard sample was mixed with the egg yolk IgY solution at a ratio of 7:1 (*v*:*v*) and the mixture was incubated at 42 °C for 1.5 h; the sample mixed with saline served as a control. Upon completion of incubation, the mixtures were subjected to ELISA analysis for assessing the specific IgY titer as described above.

In the second assay, fresh gizzard contents from 5 euthanized broilers at the age of 6 weeks were collected, pooled, and immediately used for a stability assay. The fresh gizzard content sample was gently mixed with same volume of saline and centrifuged at 5000× *g* for 5 min. The pH of supernatant was measured. Subsequently, the gizzard supernatant and HCl solution (1 mM, pH 3.0) were mixed with the egg yolk IgY solution at a ratio of 20:1 (*v*:*v*), respectively, and the mixture was incubated at 42 °C for 30 or 60 min; the sample mixed with saline served as a control. Upon completion of incubation, the mixtures were subjected to ELISA analysis for assessing the specific IgY titer as described above.

### 2.6. Statistical Analysis

Statistical analyses were conducted using Mann–Whitney test in GraphPad Prism 8. Data, unless specified, were presented as mean ± standard deviation (SD). A probability level of *p* < 0.05 was considered a statistically significant difference.

## 3. Results

### 3.1. The Specific Egg Yolk IgY Was Highly Stable in Chicken Feed

ELISA analysis was performed to examine the stability of the Ent conjugate vaccine-specific IgY in the daily supplied feed containing hyperimmune egg yolk powder. As shown in Figure 1A, the specific IgY titer in the newly prepared feed that was supplemented with 2% of egg yolk powder (0 d) was 14 log_2_ units higher than that in the control feed. After 2 days of exposure to room temperature at more than 32 °C in the chicken house, the specific IgY (2 d) maintained a high level, more than 8000-fold higher than in the control feed (Figure 1A).

### 3.2. The Anti-Ent Egg Yolk Failed to Reduce Intestinal Colonization of C. jejuni

As shown in Figure 1B,C, supplementation of chicken feed with the anti-Ent egg yolk powder did not significantly reduce *C. jejuni* colonization in the chicken intestine when compared to the control feed. Specifically, as expected, all of the five randomly selected chickens in each group before *C. jejuni* challenge (0 DPI) were *C. jejuni* free in both the cecum and ileum (out of detection limit) (Figure 1B,C). At 4 days following *C. jejuni* challenge (4 DPI), the chickens in both groups displayed a high level of *C. jejuni* colonization in the cecum; interestingly, the *C. jejuni* level in the Ent group (9.38 log_10_ CFU/g) was even higher (*p* < 0.05) than in the control group (8.46 log_10_ CFU/g) (Figure 1B). By 9 DPI, the anti-Ent egg yolk still did not exert significant inhibition on *C. jejuni* colonization in the cecum (Figure 1B). As expected, for the chickens within the same group, the level of *C. jejuni* in the ileum was significantly lower (*p* < 0.05) than in the counterpart cecum at 4 or 9 DPI (Figure 1B,C). Similar to the findings in the cecum, there was no significant difference in *C. jejuni* colonization in the ileum between the control and Ent groups (Figure 1C).

### 3.3. Cecum and Ileum Microbiota Was Not Dramatically Affected by the Anti-Ent Egg Yolk

To examine the changes of intestinal microbiota in response to anti-Ent egg yolk treatment, 16S rRNA gene amplicon sequencing was conducted. Regarding Shannon diversity (Figure 2A), the microbiota exhibited similar diversity between control and Ent groups in both the cecum (*p* = 0.06) and ileum (*p* = 0.60). To compare the microbial structure among different samples, PCoA was performed based on Bray–Curtis distance. As shown in Figure 2B, the dots of cecum microbiota in the control group were more clustered than those in the Ent group (0.51 ± 0.16 vs. 0.58 ± 0.20), indicating significant dissimilarity in the structure of cecum microbiota between the two groups (*p* < 0.01). The dots of ileum microbiota in the control or Ent groups were similarly scattered (0.57 ± 0.24 vs. 0.62 ± 0.31), meaning no significant difference in their microbiota structure (*p* = 0.18). We also looked at the specific taxonomic composition of intestinal microbiota in each group (Figure 2C). The families of Lachnospiraceae and Enterococcaceae affiliated to the phylum Firmicutes were predominant in the cecum and ileum, respectively. The second abundant phylum was Proteobacteria (mainly the family Enterobacteriaceae) (Figure 2C). Generally, there was no dramatic difference in taxonomic composition between the control and Ent groups at different time points (Figure 2C). In addition, there was a notable expansion of Campylobacteraceae (the family *C. jejuni* belongs to) in both treatment groups at days 9 and 14 (i.e., 4 and 9 DPI) in the cecum but not ileum (Figure 2C). Taken together, microbiota analysis revealed there was no dramatic change in the intestinal bacterial community caused by anti-Ent egg yolk.

### 3.4. The Levels of the Specific Egg Yolk IgY Were Not Significantly Different between Treatment Groups

To properly interpret the findings resulting from oral administration of hyperimmune egg yolk powder, it was critically important to examine the level and bioavailability of the specific IgY in different GI segments as well as the interlinked circulating system. Therefore, in this study, ELISA analysis was further performed to determine the titer of the Ent conjugate-specific IgY in different GI contents (cecum, ileum, duodenum, and gizzard) as well as serum samples collected from individual chickens. We speculated that the titer of the specific IgY in the chickens receiving hyperimmune egg-yolk-powder-supplemented feed should be significantly higher than in the chickens receiving the control feed, although the magnitude of difference might vary with respect to different GI segments.

However, despite the exceptionally high level of the specific egg IgY in the treatment feed (Figure 1A), unexpectedly, the levels of the specific egg yolk IgY in the cecum (Figure 3A), ileum (Figure 3B), duodenum (Figure 3C), and gizzard (Figure 3D) of the Ent group were consistently low or below the detection limit at different time points, similar to those observed in the control group (Figure 3A–D). Although serum samples displayed a high basal level of the specific IgY titer, there was no significant difference in the titer of the specific egg yolk IgY in chicken serum between control and Ent groups on all three sampling days (Figure 3E). These surprising findings strongly suggested that the ingested specific egg yolk IgY was not stable when passing through the GI tract, particularly the gizzard.

### 3.5. Ex Vivo Assays Revealed Instability of the Egg Yolk IgY in Gizzard

To evaluate the stability of the egg yolk IgY in the gizzard and duodenum, we collected gizzard and duodenum contents from two independent sources of broilers (6–7 weeks old) and conducted ex vivo assays.

In the first assay, treatment of the hyperimmune egg yolk IgY in duodenum content at 42 °C for 2.5 h led to the same high titer of the specific IgY as in the saline control, indicating that the egg yolk IgY was stable in the duodenum content even after a long time of treatment (Figure 4A). Meanwhile, the duodenum content without the addition of the egg yolk IgY (negative control) displayed a very low basal level of the specific IgY (Figure 4A). For the ex vivo stability assay using gizzard content (Figure 4B), when compared to the saline control that displayed the high level of the IgY titer, the gizzard content with pH 3.0 showed a dramatically reduced titer of the specific IgY that was below the detection limit. However, when the pH of the gizzard content was adjusted to 7.0, the IgY titer was as high as observed in the saline control (Figure 4B). In addition, the titer of the specific IgY in the gizzard content (pH either 3.0 or 7.0) without the addition of the egg yolk IgY (negative controls) was under the detection limit, indicating the gizzard content had extremely low background noise for measuring the specific egg yolk IgY using ELISA analysis (Figure 4B). Collectively, the findings from the first ex vivo assay indicated that the egg yolk IgY was very unstable in the chicken gizzard, which was dependent on the acidic environment in the gizzard.

The first assay had three potential limitations. First, the intestinal contents were obtained from a distant state via overnight shipping, which might have affected sample quality for the ex vivo assay. Second, the pH of the gizzard content extract was purposely adjusted to 3.0 to mimic average pH condition in the chicken gizzard. However, using the original saline-extracted gizzard content (with a slightly higher pH than 3.0) for the stability assay would provide an additional clue to explain the instability of IgY in the gizzard. Finally, although the instability of the egg yolk IgY was dependent on low pH, it was still not clear if this was due to the direct impact (i.e., denaturation of the IgY) or indirect impact (i.e., low pH-dependent activity of pepsin, a proteolytic enzyme in gizzard). To address these issues, another assay was performed using freshly prepared gizzard contents with additional controls and time points (Figure 4C). Consistent with the findings in the first assay (Figure 4B), incubation of the egg yolk IgY in the gizzard content extract with a natural pH 3.3 for as short as 30 min led to a 32-fold reduction in the specific IgY titer when compared to the saline control (Figure 4C). Notably, incubation of the egg yolk IgY with a hydrochloric acid solution (pH 3.0) for up to 60 min did not affect the titer of the specific IgY at all, indicating acidic denaturation did not contribute to the instability of the IgY in the gizzard content (Figure 4C).

## 4. Discussion

Passive immunization with specific egg yolk antibodies is emerging as a promising alternative to antibiotics for the treatment and prevention of various animal and human diseases [1,2]. In particular, hyperimmune egg yolk IgY has been commonly administrated orally through feed or drinking water to prevent and control enteric diseases in food animals [1,16]. Our recent study [13] demonstrated that immunization of chickens with the novel Ent conjugate vaccine significantly reduced *C. jejuni* colonization in the intestine. With the aid of the production of a high titer of anti-Ent egg yolk IgY [14], in this study, we examined the efficacy of oral administration of the anti-Ent egg yolk IgY for *C. jejuni* control in broiler chickens. Surprisingly, we did not observe any protection conferred by the anti-Ent egg yolk IgY against *C. jejuni* colonization in the intestine. Subsequently, we were more surprised at the finding that the specific IgY was barely detected in different sections of the GI tract, including the upper digestive organ gizzard. Notably, to our best knowledge, none of the previous animal trials evaluating the efficacy of egg yolk IgY has examined the level and bioavailability of egg yolk IgY in the GI tract following ingestion. Given the harsh environment encountered by egg yolk IgY in the GI tract (e.g., low pH and proteases), the in vivo availability of egg yolk IgY is a key issue that needs to be addressed for data interpretation and understanding the mechanism of the passive immunity conferred by egg yolk IgY. This study, for the first time, utilized ELISA analysis to directly quantitatively measure the specific egg yolk IgY level along different sites of the GI tract as well as in systemic circulation (Figure 3). Examining the level of the specific egg yolk IgY in serum can provide insights into the potential absorption of egg yolk IgY into systemic circulation through intestinal epithelia and justify oral administration of egg yolk IgY for passive immune protection against systemic infections.

To confirm the in vivo finding—instability of IgY in the GI tract—and to better understand the fate of egg yolk IgY in a chicken’s GI tract following ingestion, we also performed ex vivo assays to examine the stability of the egg yolk IgY in both the gizzard and duodenum contents in this study. The gizzard and duodenum are two major digestive organs in chickens. In the gizzard, the pepsin under the acidic condition maintained by secreted hydrochloric acid, together with grinding activity, plays an important role in the initial digestion of feed with about 60 min of retention [25]. The duodenum, the upper digestive organ immediately after the gizzard, contains digestive enzymes, such as the trypsin released from the pancreas [26]. Using a well-controlled system plus different sources of samples, our ex vivo assays provided compelling evidence demonstrating that the gizzard environment dramatically destabilized the egg yolk IgY, which was likely mediated through potent enzymatic degradation by the pepsin (Figure 4). Using a hydrochloric acid solution (pH 3.0) as a control (Figure 4C), we ruled out the possibility that acidic condition might denature egg yolk IgY and/or affect the antigen-antibody interaction during ELISA analysis, which might have led to the observed low IgY titer in the gizzard content. It has been reported that chicken IgY was resistant to bovine trypsin and chymotrypsin treatment but could be degraded in swine pepsin [27]. Notably, the functionality of chicken pepsin strictly relies on a low pH with optimal activity at pH 2.8 [28,29], which may explain why the magnitude of the IgY titer reduction in the original gizzard extract (pH 3.3) (Figure 4C) was lower than in the extract adjusted to pH 3.0 (Figure 4B). At this stage, it is still unknown if chicken age can influence the instability of IgY in the gizzard. In addition, direct evidence showing degradation of egg yolk IgY by the pepsin in the chicken gizzard is still lacking. To address these issues, a comprehensive study using various individual samples in conjunction with molecular and biochemical approaches is highly warranted in the future.

Based on the findings from this study, potential use of hyperimmune egg yolk powder to control enteric diseases should be re-evaluated in poultry and other food animals in the future. In particular, the in vivo stability of egg yolk IgY needs to be investigated in parallel to other phenotype assessments. In addition, to increase stability and bioavailability of administrated egg yolk IgY in the GI tract, protective coating of the IgY is a feasible strategy. Previously, methacrylic acid copolymer, chitosan-alginate, and alginate/carrageenan hydrogels have been utilized for the encapsulation of specific egg yolk IgY, which, to different extents, prevented activity loss of the IgY and achieved controlled releases of the IgY in a simulated GI fluid [30,31,32]. In the future, cost-effective encapsulation methods should be developed for site-specific delivery and controlled release of innovative hyperimmune egg yolk IgY used in livestock, such as the anti-Ent egg yolk IgY investigated in this study.

Although passive immunization with the anti-Ent hyperimmune egg yolk IgY failed to reduce intestinal colonization of *C. jejuni*, we will continue to explore practical and cost-effective vaccination routes for the Ent conjugate vaccine to control Gram-negative pathogens in poultry. For instance, in ovo vaccination, especially with the adoption of an automatic in ovo injection system [33], is an attractive approach for the poultry industry to control infectious diseases [34]. In addition, mucosal vaccination through the oculo-nasal route or via spray is a potentially convenient approach to induce the desired local mucosal immunity (e.g., respiratory or GI tract) in poultry and other food animals [35]. Given that the pathogens targeted by the Ent conjugate vaccine are mainly involved in mucosal infections (e.g., avian pathogenic *Escherichia coli* in the respiratory tract [36]; *C. jejuni* and *Salmonella enterica* in the intestinal tract [37,38]), this strategy, with proper adjuvants and delivery systems [39], would also expand the application of the Ent conjugate vaccine in food animals.

## 5. Conclusions

In this study, oral administration of broiler chickens with the anti-Ent egg yolk powder failed to confer protection against intestinal colonization of *C. jejuni*, which was due to the instability of the IgY in the gizzard. This study, for the first time, assessed the availability and stability of the hyperimmune egg yolk IgY in different segments of the chicken GI tract using both in vivo and ex vivo systems.

## Figures and Tables

**Figure 1 vaccines-09-00569-f001:**
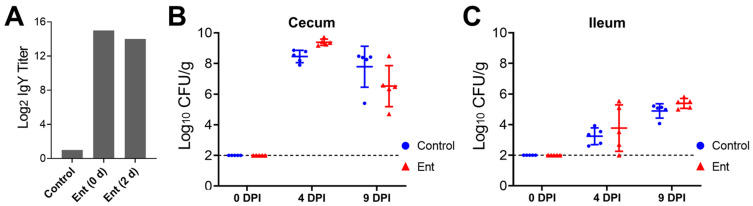
*C. jejuni* colonization levels in the chicken intestine upon anti-Ent egg yolk powder treatment. (**A**) Ent conjugate vaccine-specific IgY titer in chicken feed. ELISA analysis was performed to determine the titer of the Ent conjugate-specific egg yolk IgY in the control feed (regular feed) or the feed supplemented with 2% (*w*/*w*) hyperimmune egg yolk powder (designated as “Ent”). With respect to the feed supplemented with the egg yolk powder, both freshly prepared feed (0 d) and the feed with 2 days of exposure in the chicken house (2 d) were subjected to ELISA analysis. Each bar represents duplicate measurements of the sample. (**B**) Colonization of *C. jejuni* NCTC 11168 in the cecum. (**C**) Colonization of *C. jejuni* NCTC 11168 in the ileum. The intestinal samples were collected from 0, 4, and 9 DPI for *C. jejuni* enumeration. Each dot represents log_10_-transformed CFU per gram of intestinal contents in an individual chicken. The bar represents mean ± SD within each treatment group. The dashed line indicates the detection limit.

**Figure 2 vaccines-09-00569-f002:**
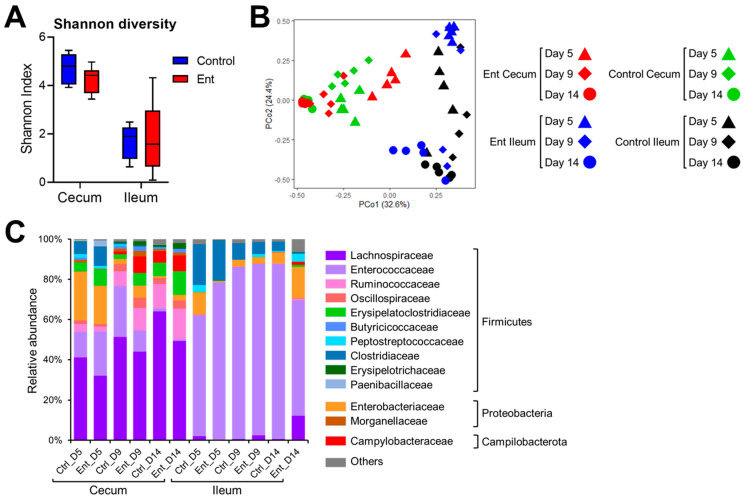
Analysis of intestinal microbiota in chickens using 16S rRNA gene amplicon sequencing. (**A**) Shannon diversity of bacterial community in the cecum and ileum. (**B**) Principal coordinate analysis, based on Bray–Curtis distance, on microbiota structure in the cecum and ileum. (**C**) Relative abundance of specific family/phylum in the cecum or ileum of the chickens in two groups at different time points. Each bar represents the averaged abundance of five samples within each group.

**Figure 3 vaccines-09-00569-f003:**
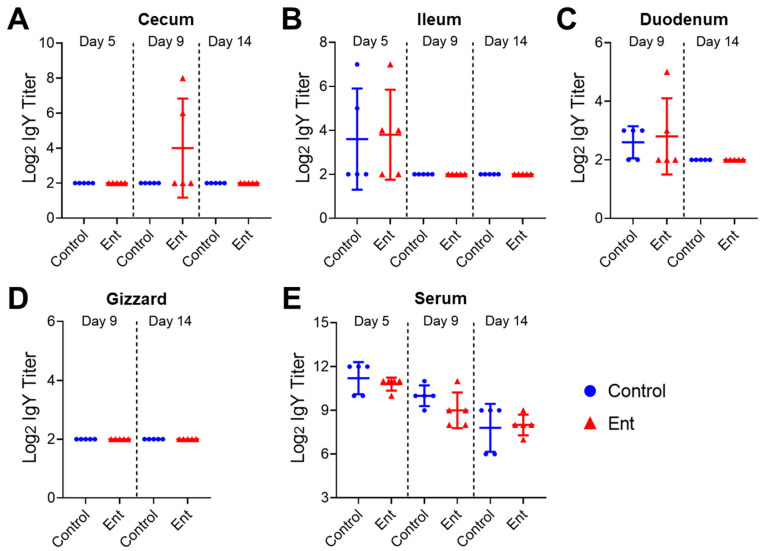
The levels of the specific egg yolk IgY in different GI contents and serum. ELISA analysis was performed to determine the titers of the Ent conjugate-specific IgY in the (**A**) cecum, (**B**) ileum, (**C**) duodenum, (**D**) gizzard, and (**E**) serum collected from individual chickens fed with control or Ent feeds at different time points. Each dot represents duplicate measurements of a specific sample. Each bar represents mean ± SD within each group.

**Figure 4 vaccines-09-00569-f004:**
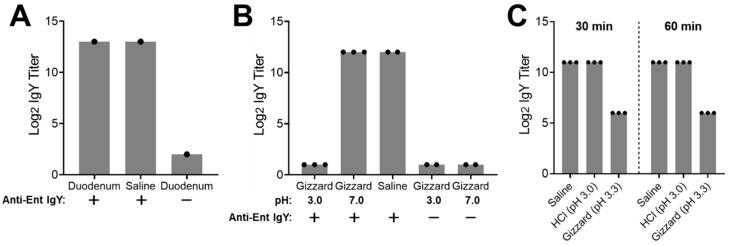
Ex vivo stability of the specific egg yolk IgY in duodenum and gizzard contents. (**A**) Ex vivo stability of the specific egg yolk IgY in the duodenum. The duodenum content was spiked with the hyperimmune egg yolk IgY and incubated at 42 °C for 2.5 h, followed by ELISA analysis of the specific IgY titer. The saline spiked with the hyperimmune egg yolk IgY served as a positive control while the duodenum content without addition of the egg yolk IgY served as a negative control (detailed in Materials and Methods). (**B**) Ex vivo stability of the specific egg yolk IgY in the gizzard. The gizzard content (pH 3.0 or 7.0) was spiked with the hyperimmune egg yolk IgY and incubated at 42 °C for 1.5 h, followed by ELISA analysis of the specific IgY titer. The saline spiked with the hyperimmune egg yolk IgY served as a positive control while the gizzard contents without addition of the egg yolk IgY served as negative controls. (**C**) Effect of pH and incubation time on ex vivo stability of the specific egg yolk IgY in the gizzard. Fresh gizzard content was collected and immediately subjected to an ex vivo stability assay. The freshly prepared gizzard content extract (pH 3.3) was spiked with the hyperimmune egg yolk IgY and incubated at 42 °C for 30 or 60 min, followed by ELISA analysis of the specific IgY titer. The hydrochloric acid (pH 3.0) and saline that were spiked with the egg yolk IgY served as controls. Each dot represents mean of duplicate measurements of an independent sample. Each bar represents mean ± SD of the indicated number of samples.

## Data Availability

The data presented in this study are available on request from the corresponding author.

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
