# Peer review of "Passive Immunization of Chickens with Anti-Enterobactin Egg Yolk Powder for Campylobacter Control"

_vaccines, 2021, doi:10.3390/vaccines9060569_

Round 1

Reviewer 1 Report

This is a well-written manuscript regarding a novel measures to control Campylobacter infection by passive vaccination strategy.

Author Response

We appreciate the reviewer for the positive comment on this manuscript.

Reviewer 2 Report

Passive immunization of chickens with anti-enterobactin egg yolk powder for campylobacter control 

Dear author and editor:

The article talked about passive immunization with anti-enterobactin antibody to verify its potential activity against intestinal colonization. The author found that the anti-enterobactin egg yolk powder failed to give protection against campylobacter because of IgY instability in gizzard. 

The article could be published in vaccine after a minor revision. 

I have some comments on it: 

  • Why the author did not evaluate the activity of the egg yolk powder against campylobacter in another animal model. 
  • The encapsulation of egg yolk IgY is a good idea for specific delivery and protection. What do you think about using hydrogels as a drug delivering method to achieve a controlled therapeutic delivery to a specific site in gastrointestinal tract and protect antibody from gastric condition. The author can discuss this point. 
  • Do you think that anti-enterobactin egg yolk IgY could be more effective against systemic infection? What is the advantage and disadvantage from using these antibodies?

Thank you very much, best regards 

Reviewer 3 Report

General overview of the manuscript

This manuscript is providing important information on the use of Ent conjugate-specific egg yolk IgY to control Campylobacter intestinal colonisation in broiler chickens of which its applicability could solve problems as far as food safety is concerned. Despite previous literature to narrate on the success of using this chicken egg product in controlling Campylobacter colonisation in chickens through injection route, its applicability was not cost-effect which lead to designing the trial study on administering the freeze-dried anti-Ent egg yolk powder through the oral route which could have been cost-effective and easily applicable. The results of this trial were measured by determining the level of Ent conjugate vaccine-specific IgY titer in contents from different sections of the GIT of the sacrificed trial chickens at different time intervals post-Campylobacter oral gavage and daily feeding with anti-Ent egg yolk powder-supplemented feed. There was no significant difference in the Ent conjugate-specific egg yolk IgY titre between the treated and control group indicating low protection capability of the test components when administered orally. In addition, the stability of the test materials was tested in vitro to determine its stability in GIT contents, the trial which showed pH-independent instability of the test materials along with the GIT, especially in the gizzard. Also, test materials did not cause any significant changes in GIT microbiota in terms of diversity, abundance, and type.

This manuscript is well designed and provides important information on the use of these materials as the means of controlling Campylobacter colonisation and despite the materials not showing promising results, the authors proposed different options including coating the test materials which shows the avenue for future study. However, there are some areas that need authors’ attention for improvement.

Introduction section

The introduction provides a good summary of what has been done so far in using the Ent conjugate-specific egg yolk IgY to control GI bacterial colonisation and the hypothesis is well stated, however, this section can be further improved to include more useful information. Since the manuscript is testing the materials for controlling bacterial colonisation, I expected the manuscript to talk about (in nutshell) the factors affecting the Campylobacter colonisation in chickens. Also, more information is required on the importance of Campylobacter in food safety Globally i.e. annual morbidity and mortality.

Materials and Methods section

Line 148: State primer and genomic DNA concentration

Line 206-2017: There is nowhere in the material and method section explaining how the Ent conjugate vaccine-specific IgY titer in chicken feed was determined.

Figure 1: I expected to see a zero titer of Ent conjugate-specific egg yolk IgY in the control feed (regular feed), what could be the source of Ent conjugate-specific egg yolk IgY observed in the control feed?

Figure 1: There is a slight drop in the level of Ent conjugate-specific egg yolk IgY in feed on day 0 and 2, does storage time has something to do with the level of the titre in the feed?

Line 234 – 235: Why did you expect the colonisation level to be lower in illium than in caecum?

Line 241 – 243: Show clearly which value is for the control and treatment group, you need to re-write this sentence.

Figure 3: The level in the GIT drops drastically but in the serum, there is a sustained level of Ig Y titer upon oral administration, if I may ask, can the high titer in the serum prevent the colonisation in the GIT?

Line 255 – 257: In addition, there was a notable expansion of Campylobacteraceae in both treatment groups on days 9 and 14 (i.e.,4 and 9 DPI) in cecum but not ileum (Figure 2C). Can you account for this in the discussion?

Line 336: ……..it was still not clear it was due to direct impact…………. Re-write this sentence segment

Discussion section

The discussion is well written but lacks some explanation in some of the issues encountered as the results of the trial. In spite of the low titre in GIT contents obtained from different sections of the GIT contents, there was a sustained titre level in the serum at different days post-inoculation, I expected the authors to provide their opinion on the importance of this sustained titre in the serum in relation to the prevention of bacterial colonisation in the GIT. There was a notable expansion of Campylobacteraceae in both treatment groups at days 9 and 14 (i.e.,4 and 9 DPI) in cecum but not ileum (Figure 2C), I expect this to be accounted for in the discussion. In addition, the authors remained silent about the chickens within the same group having a higher level of C. jejuni in the caecum than in ileum at 4 or 9 DPI. Finally, the discussion misses some concluding remarks.

Line 361 – 364: This sentence is not clear.

Reviewer 4 Report

Overall, the manuscript is very interesting, it presents an overview on passive immunization of chickens with anti-enterobactin egg yolk powder for Campylobacter control. The study is well-presented, methods are clear and reliable and in some aspect innovative. Data are presented in an appropriate way and figures are adequately detailed. Paper is a useful addition to the data in this area.

Author Response

We appreciate the reviewer for the positive comments on this manuscript.

Reviewer 5 Report

General comments

This study is very relevant in that the incidence of Campylobacter infection originated from animals has been increasing recently across the world and antibiotic resistant Campy is considered as a serious public health threat. Thus, developing non-antibiotic control strategies are highly encouraged. In this light, anti-Ent Campy vaccine and its oral administrative approach is an innovative work. In addition, I appreciate in this paper that negative results (contrary to the hypothesis) are reported, which most of the time are under reported.

This paper is well-written, and the study design is scientifically sound. A little concern I have is the statistical method used, I would check the difference between groups using non-parametric tests for such small size of animal studies (5 per group at each sampling time point).

Specific comments

Introduction

  • Background and the rationale of the study were well explained.
  • Clear hypothesis and objective.

M & M

  • Excellent
  • The statistical test might need a second look at it. For a sample size of 5 in each group, I would use non-parametric t-tests (Mann-Whitney or Wilcoxon signed-rank test).

Results

  • Excellent
  • 2.B visually it looks like there is no significant difference between the microbial structure of ENT and Control cecum. Is the p-value reported here is an adjusted one?

Discussion

  • The findings were discussed in the light of previous studies and the way forward to deliver anti-Ent vaccine was clearly stated.
  • Could nano technology be a potential means of delivering this vaccine orally?
